# Genomic Analysis of *Clostridium perfringens* BEC/CPILE-Positive, Toxinotype D and E Strains Isolated from Healthy Children

**DOI:** 10.3390/toxins11090543

**Published:** 2019-09-19

**Authors:** Raymond Kiu, Kathleen Sim, Alex Shaw, Emma Cornwell, Derek Pickard, J. Simon Kroll, Lindsay J. Hall

**Affiliations:** 1Gut Microbes & Health, Quadram Institute Bioscience, Norwich NR4 7UQ, UK; Raymond.Kiu@quadram.ac.uk; 2Faculty of Medicine, Imperial College London, London W2 1NY, UK; K.Sim@imperial.ac.uk (K.S.); a.shaw@imperial.ac.uk (A.S.); emma.zabaleta@hotmail.com (E.C.); s.kroll@imperial.ac.uk (J.S.K.); 3Department of Medicine, University of Cambridge, Cambridge CB2 0QQ, UK; djjp2@medschl.cam.ac.uk

**Keywords:** *Clostridium perfringens*, epsilon toxin, iota toxin, whole genome sequencing, infants, binary toxins, genome analysis

## Abstract

*Clostridium perfringens* toxinotype D, toxinotype E, and gastroenteritis-linked BEC/CPILE-positive strains have never been reported in healthy children. We isolated, whole-genome sequenced and bioinformatically characterised three *C. perfringens* isolates—type D (IQ1), type E (IQ2) and BEC/CPILE-positive (IQ3), recovered from the stools of three healthy two-year-olds, which were further compared to 128 *C. perfringens* genomes available from NCBI. The analysis uncovered a previously under-described putative toxin gene *alv* (alveolysin) encoded by isolates IQ2 and IQ3, which appeared to be a clade-specific trait associated with strains from domestic animals. A plasmid analysis indicated that the iota-toxin was encoded on a near-intact previously described plasmid pCPPB-1 in type E strain IQ2. The BEC genes *becA* and *becB* were carried on a near-identical pCPOS-1 plasmid previously associated with Japanese gastroenteritis outbreaks. Furthermore, a close phylogenetic relatedness was inferred between the French *C. perfringens* type E isolates cp515.17 and newly sequenced IQ2, suggesting geographical links. This study describes novel *C. perfringens* isolates from healthy individuals which encode important toxin genes, indicating the potential spread of these veterinary and clinically important strains and mobile genetic elements, and highlights areas for future research.

## 1. Introduction

*Clostridium perfringens* is a fast-growing Gram positive, anaerobic and spore-forming pathogen known to produce >20 toxins. These toxins induce various disease states in both humans and animals including gastroenteritis, gas gangrene, necrotic enteritis, necrotising enterocolitis, and enterotoxaemia [1,2,3,4]. Typically, based on the combinations of typing toxins that are produced, including alpha toxin, beta toxin, epsilon toxin and iota toxin, *C. perfringens* strains were classified into 5 distinct toxinotypes. More recently, this typing scheme has been updated to 7 toxinotypes (including NetB and enterotoxin CPE) to better reflect clinical relevance (toxinotypes A–G) [5,6].

*C. perfringens* toxinotype D isolates, capable of producing alpha toxin (encoded by *plc*) and epsilon toxin (*etx*), are known to cause natural enterotoxaemia (also commonly known as ‘pulpy kidney’) primarily in sheep, goats, and occasionally cattle [7,8,9]. Enterotoxaemia in sheep is characterised by sudden death syndrome, brain lesions, lung oedema and enterocolitis. The absorption of the epsilon toxin followed by bloodstream circulation to internal organs such as the brain and kidney results in an elevated blood pressure and neurological disease [10,11,12,13]. Epsilon toxin is also known as the third-most potent toxin after the botulinum and tetanus toxins [14], with the epsilon toxin gene *etx* typically encoded on plasmids [15].

*C. perfringens* type E isolates can produce alpha toxin and iota toxin (encoded by the *iap* and *ibp* genes) and may also carry secondary toxin genes such as *cpb2* (beta2 toxin). These strains are primarily associated with diseases in domestic animals including enterotoxaemias in rabbits, and enteritis in lambs and cattle [16,17]. The binary iota toxin genes *iap* (encodes the enzymatic component) and *ibp* (the binding component) are both plasmid-encoded [18].

More recently, it was reported that a potent binary enterotoxin of *C. perfringens* (BEC; also, CPILE), encoded by *becA* and *becB* genes, was linked to food poisoning outbreaks in Japan, with its enterotoxic activity confirmed experimentally [19,20]. This enterotoxin, distinct from typical gastroenteritis-associated enterotoxin CPE, as found to be carried on a ~54 kb plasmid, suggesting its potential for horizontal gene transfer [19].

In this study, we performed comparative genomics to analyse *C. perfringens* strains obtained from healthy two-year-old children who had been born prematurely. Two of the isolates were characterised as the ‘typical’ veterinary-associated toxinotypes D (IQ1) and E (IQ2), respectively, and one was a BEC-positive strain (IQ3). This study contributes further understanding of toxin genes encoded in *C. perfringens* isolates from otherwise healthy individuals, including epsilon toxin, iota toxin and perfringolysin O. Furthermore, the type E and BEC-positive strains were also found to carry a rarely-discussed toxin—alveolysin—which despite sharing a high homology (~86% nucleotide identity) with perfringolysin O, had distinct genetic sequences. A further in-depth analysis on toxin-encoding plasmids in toxinotype D, toxinotype E and BEC-positive *C. perfringens* strains provided additional insights into the phylogenetic relatedness of these strains.

## 2. Results

### 2.1. Genomic Description of Newly Sequenced C. perfringens Isolates

In this study, we isolated and sequenced three *C. perfringens* strains from preterm-born healthy children who were originally recruited as part of the two-year NeoM (Neonatal Microbiota) study (Appendix A and Table 1). These three isolates were sequenced at an ultra-high coverage (195×–212×), which enabled us to obtain high quality low-contig short-read sequenced draft genome assemblies at a high N50; IQ2 (15 contigs, N50: ~2.1 mb), IQ3 (20 contigs, N50: ~2.2 mb) and IQ1 (41 contigs, N50: ~450 kb). These genomes were taxonomically confirmed to be *C. perfringens* at a species level using various genomic approaches, including the 16S rRNA gene, Average Nucleotide Identity (ANI) and digital DNA-DNA Hybridisation (dDDH). The GC percentages (%) are stable between the genomes (range: 27.98%–28.03%) and were predicted to contain an average of ~3272 genes per genome. These three *C. perfringens* strains were toxinotyped as type D (IQ1; encoding epsilon toxin), type E (IQ2; encoding iota toxin) and type A (IQ3; carrying BEC) respectively. A bioinformatics analysis of antimicrobial genes indicated the presence of only *tetA(P)* in isolates IQ2 and IQ3.

### 2.2. Comparisons of Toxin Gene Sequences

Several important toxin gene sequences were compared, including; *etx* (epsilon-toxin), *iap* (iota-toxin component a), *ibp* (iota-toxin component b), *becA* (binary enterotoxin component a), *becB* (binary enterotoxin component b), *pfoA* (thiol-activated toxin perfringolysin O) and *alv* (thiol-activated toxin alveolysin), as demonstrated in Table 2, Table 3 and Table 4.

Epsilon-toxin gene *etx* variants appeared to be well conserved among 5 different *C. perfringens* strains at the nucleotide level, with only one different nucleotide identified at position 726 (987 bp alignment), which did not alter the amino acid residue serine (S) at position 254 (Table 2). The binary iota-toxin gene components *iap* and *ibp* were investigated in 4 toxinotype E strains, including the recently published French strains cp508.17 and cp515.17 [21]. Iota-toxin gene variants were shown to be highly conserved at a nucleotide identity >99.85% (only one variable amino acid at position 134, either isoleucine or threonine) among strains cp508.17, cp515.17 and IQ2 (Table 3). Notably, strain JGS1987 had more dissimilar *iap* and *ibp* components, assuming functional changes at the amino acid level (~142 amino acid residue differences compared to IQ2; Appendix A). The BEC-toxin gene components *becA* and *becB* were identical (100% identity) to the reference *becAB* genes (Appendix A).

A thiol-activated membrane-damaging toxin alveolysin gene *alv*, first described in *Paenibacillus alvei*, was detected in the *C. perfringens* isolates IQ2 (type E) and IQ3 (BEC-positive; Appendix A) [22]. This gene is 1578 bp long, with the alveolysin protein comprising 528 amino acids. At a nucleotide level, it shares 99.62% similarity (IQ3 *alv*) with a similar gene in JP838 (accession CP010994.1; NCBI NR database) at 100% query coverage, annotated as alveolysin (nucleotide identity: 100%). In addition, this gene is similar to, but distinct (~86% identity) from, typical perfringolysin O gene *pfoA* variants in *C. perfringens* at the nucleotide level (PfoA is a frequently-detected thiol-activated toxin produced by *C. perfringens*) (Table 4). A protein sequence search (blastp) on the NCBI non-redundant protein sequences database showed that *alv* shared a sequence homology with *alv* genes in several Gram-positive bacteria, including *Paenibacillus alvei* (~74%), *Paeniclostridium sordellii* (~78%), *Paraclostridium bifermentans* (~77%), *Clostridium argentinense* (~78%), and *Bacillus luciferensis* (~70%).

### 2.3. Comparative Study of Toxin-Encoding Plasmids

Key *C. perfringens*-derived toxin genes are known to be carried on plasmids, including epsilon toxin, iota toxin and BEC [19,23,24]. Thus, we attempted to computationally reconstruct and investigate toxin-encoding plasmids using both a reference-based assembly (based on NCBI plasmid sequences) and *de novo* assembly-graph approaches (by extracting connected circular contigs from assembly graphs that represent potential plasmid sequences). Plasmid replication genes (*ori*) were also confirmed in these extracted sequences to validate our investigations (Figure 1).

Initially, pIQ1a was assembled in 13 contigs; length of 53,937 bases out of 64,753 bases based upon the only *etx*-reference plasmid pCP8533etx (from a lamb dysentery type B isolate in 1953; NCTC8533) available in the NCBI nucleotide database [25]. This suggested that pIQ1a was not the same as pCP8533etx (Figure 1A; Table 5), which was further supported by an unbiased assembly-graph approach (based on comparing the contig connection and sequencing depths: ~188X in plasmids vs. < 50X in chromosomes). This analysis predicted that the circular plasmid pIQ1b was smaller (~48 kb) than pCP8533etx and encoded *etx,* but not *cpb2* as in pCP8533etx. This finding agrees with a previous study that reported that the *etx*-plasmid in *cpe*-negative *cpb2*-negative type D *C. perfringens* strains was typically ~48 kb long, indicating that the *etx*-plasmid pIQ1b in strain pIQ1 is a typical *cpe*-negative *cpb2*-negative type D plasmid [15].

Near-identical toxin-encoding plasmids were identified in both strains IQ2 and IQ3 with respect to known *C. perfringens* plasmids (Table 5; Figure 1B,C). The predicted conjugative plasmid pIQ2 was highly similar to the reference plasmid pCPPB-1 (originally isolated from retail meat products in Japan); ~67 kb in length, assembled into one full contig (GC ~26%), and encoding both iota toxin gene components and the enterotoxin gene *cpe* [24]. The predicted plasmid pIQ3 appeared near-identical to the reference plasmid pCP-OS1(originally from a clinical *C. perfringens* strain isolated from a diarrhoeal patient in Osaka, Japan) at a plasmid length ~54 kb, GC content ~25% and assembled into one full contig, carrying intact *becAB* genes [19].

### 2.4. Phylogenetic Analysis

To understand the phylogenetic positions of these novel isolates, we constructed a core-genome based phylogenetic tree of 132 *C. perfringens* genomes, including all available 128 *C. perfringens* genome assemblies from the NCBI RefSeq database (July 2019). The type E isolate IQ2 was shown to be closely related, although not identical, to the French type E strains cp515.17 and cp508.17 (pairwise genetic distance: 649 SNPs, Figure 2). The BEC-positive isolate IQ3 was also found within the same clade as the type E isolates, which may indicate a geographical relatedness. Furthermore, the type D isolate IQ1, which carries *etx,* was located within a *C. perfringens* clade that comprises the type strain ATCC13124, and other type D strains including JGS1721 and NCTC8503.

Notably, the alveolysin toxin gene *alv* was demonstrated to be clade-specific (Figure 2), and thus likely to be chromosomally encoded. This clade consisted mainly of foal- and dog-associated type F isolates that encode the pore-forming toxin gene *netF,* alongside three type E isolates: cp515.17, cp508.17, IQ2 and BEC-positive isolate IQ3 [26].

## 3. Discussion

Conventionally, *C. perfringens* toxinotype D and E isolates are associated with diseases in domestic animals, including sheep and cattle, and rarely in humans [7,18]. To our knowledge, this is the first study that has reported, sequenced and analysed *C. perfringens* toxinotype D and E strains isolated from healthy children, alongside a BEC-positive isolate. We applied bioinformatics to analyse these isolates, including understanding key toxin gene variants, toxin-encoding plasmids and phylogenetic relatedness.

Previously, four *C. perfringens* type E strains (Japan, 2011) were reported to encode iota toxin genes on a pCPPB-1 plasmid (~67 kb), which also encoded a functional *cpe* gene (strains isolated from retail meat products) [24]. We also identified a near-identical pCPPB-1 plasmid, in our *C. perfringens* type E strain IQ2, which was surprising given that the isolate was obtained from an otherwise healthy child from the UK. Both iota-toxin gene components were identical to those in the French type E isolate cp515.17 (isolated from a 60-year-old woman, subsequent to food poisoning) and similar to cp508.17 (which originated from a calf), which are both known to carry a pCPPB-1 plasmid [21]. This suggests that this specific iota-toxin encoded plasmid is widely disseminated, or that it potentially shares a common source [21]. From a disease outbreak perspective, the conjugative nature of this plasmid may allow for a transfer to other *C. perfringens* strains in the same environmental niche; however, further surveys of diseased and healthy individuals (animals and humans) coupled with experimental studies are required to probe these insights further [27]. 

More recently, BEC-positive *C. perfringens* isolates were reported in Japanese gastroenteritis outbreaks, with this binary toxin experimentally proven to be toxigenic, thus highlighting the clinical importance of this toxin in the context of public health surveillance [19]. Notably, we observed a near-identical plasmid to pCPOS-1 (from the Japanese strain OS-1), carrying identical *becAB,* in strain IQ3. This indicates that this plasmid may be more universally distributed than was first thought, which includes carriage in otherwise healthy children. Notably, a recent BEC-positive isolate obtained from a human adult was reported in Japan; however, the BEC genes *becA* and *becB* have a reduced homology, when compared to the isolate IQ3 (100%) at 98.5% and 97.9% when compared to the reported strains TS1, OS1 and W5052 [20]. The possibility of a horizontal transfer of this plasmid should be investigated as no obvious conjugative system was identified.

As there is currently a limited set of *C. perfringens* whole genome sequences, this made the phylogenetic and genome-wide profiling of type D strains that carry the deadly epsilon toxin difficult. We were not able to completely predict the epsilon-toxin encoding plasmid, in part due to the unsuccessful assembly of repeated insertion sequences in the plasmid genome. Further experimental work to complement *in silico* studies is required, particularly phenotypic validations probing the expression profiles of toxin genes.

In addition to well described toxins, we also detected alveolysin, a thiol-activated (cholesterol-binding) pore-forming toxin first identified in *Paenibacillus alvei* (a beehive-colonising bacterium). This more ‘uncommon’ toxin is known to damage cell membranes in a mechanism that resembles the homologous thiol-activated toxin perfringolysin O (theta toxin), which is typically produced by *C. perfringens* [22,28]. Importantly, membrane-damaging alveolysin also shares sequence homologies with other pore-forming toxins secreted by other Gram-positive bacteria, including listeriolysin O, pneumolysin and streptolysin O [22,29]. Despite the potential importance of alveolysin *in C. perfringens*, this toxin has received very limited research attention over the past 20 years. Its unique clade-specificity and association with canine/foal enteritis, in both type D and type E isolates, highlights the need for further research, particularly with respect to intestinal diseases in domestic animals. 

The strains described in this study were isolated as a part of a longitudinal study of the developing gastrointestinal microbiota in preterm infants [30]. Thus, it may be that the acquisition of these strains by healthy two-year-old children (residing at home) may have originally occurred during their stay in neonatal intensive care units. However, based on our larger-scale preterm *C. perfringens* screening study (unpublished data), we have not detected any other type D or E *C. perfringens* in samples collected from these infants earlier in life, or indeed in any of the other >350 preterm infants, suggesting that these strains (and plasmids) are not hospital-acquired and are not widely-circulating.

Importantly, the fact that these toxigenic *C. perfringens* isolates were obtained from healthy individuals indicated that other factors may predispose individuals to symptomatic illness. As these are gut-associated *C. perfringens* strains, the microbiota would be expected to play a significant colonisation resistance role by limiting access to nutrients and/or via direct antagonism through the production of antimicrobials. These toxigenic gut pathogens could exist within the wider microbial ecosystem without overgrowing or expressing toxin genes. However, disturbances of the resident microbiota by, e.g., antibiotics, could circumvent these protective responses, which may lead to overt disease symptoms. Further studies probing the role of the early life microbiota in this context may reveal underlying mechanisms maintaining the carriage of these potential deadly toxin types in the wider population [31,32]. 

## 4. Conclusions

Our analysis indicates the clade-specificity of the putative toxin gene *alv* (alveolysin), which was also encoded in the isolates IQ2 and IQ3. Importantly, the near-identical pCPPB-1 plasmid (previously reported in type E strains from Japan and France) was detected in the isolate IQ2, while the pCPOS-1 plasmid (previously associated with gastroenteritis outbreaks in Japan) was encoded in the isolate IQ3, highlighting a potential common source of these *C. perfringens-*derived plasmids, or otherwise a probable global distribution. Furthermore, a close phylogenetic relatedness was inferred between the French type E isolates cp515.17 and IQ2 in this study, suggesting geographical links. Whilst, WGS and a bioinformatic analysis has allowed us to probe the phylogenetic relatedness and global dissemination of three novel *C. perfringens* strains, the further large-scale sequencing of isolates from various toxinotypes is required to more accurately determine the spread of strains harboring disease-relevant toxins. Further experimental work is also required to probe toxin-disease mechanisms and complement any large-scale screening studies for the development of effective clinical and veterinary health surveillance strategies. 

## 5. Materials and Methods 

### 5.1. Ethics Declaration

The initial NeoM study (“Defining the Intestinal Microbiota in Premature Infants”) was approved by West London Research Ethics Committee Two, United Kingdom (reference number 10/H0711/39). Parents gave written informed consent for their infant to participate in the study. 

The NeoM follow-up study (“The Microbiota of the Premature Neonatal Gastrointestinal Tract: its development from birth to early childhood”) was approved by the National Research Ethics Service Committee London—Chelsea, United Kingdom (reference number 13/LO/0693). Parents gave written informed consent for their child to participate in the study. Date of approval: 2 July 2013

### 5.2. Sample Collection

NeoM follow-up study: Faecal samples were collected at home by the parents/carers of the participants and placed into a sterile faecal collection pot using the attached sterile scoop. This was posted to the laboratory ideally within 24 h and immediately aliquoted and stored in DNAase-, RNAase-free Eppendorf tubes at −80 °C upon receipt. 

### 5.3. Bacterial Isolation, Identification and Storage

Twenty-five milligrams of freshly thawed faeces were added to 0.5 mL of Robertson’s Cooked Meat Media broth. Five hundred microliters of 100% ethanol was added to the broth, vortexed for ten seconds and incubated for 30 minutes at room temperature. The broth was streaked on to fastidious anaerobic agar with 0.1% sodium taurocholate hydrate and 5% defibrinated sheep blood and incubated at 37 °C for 48 h in anaerobic conditions. Purity plates were grown for each morphologically distinct colony and for single colonies used for the species level identification by matrix-assisted laser desorption/ionization–time of flight (MALDI-TOF) using a Bruker Microflex LT (Bruker Daltonics). Stocks were stored in 70% brain heart infusion (BHI) broth (Oxoid) and 30% glycerol (Sigma) (Gillingham, UK) at −80 °C.

### 5.4. Genomic DNA Extraction and Whole Genome Sequencing

The bacterial isolates were cultured in BHI media anaerobically at 37 °C for 5–7 h prior to the DNA extraction. The genomic DNA extraction of the pure cultures was performed using a Phenol-Chloroform methodology, as described previously [33]. Sequencing was run on an Illumina HiSeq 2500 system with a paired-end insertion of length 125 bp. The sequencing coverage achieved an average of 207-fold for three novel *C. perfringens* genomes.

### 5.5. Genome Assembly and Annotation

All raw sequencing reads (FASTQ) were quality-trimmed and adapter-removed using TrimGalore v0.4.2 options -q 20 --phred33 followed by *de novo* assembly using SPAdes v3.11.1 [34]. The genome assembly was carried out using options -k 75,85,95,105 (paired-end reads insert size 2 × 125 bp) and --careful, followed by removing contigs with <300 bp using the in-house script. The assembly statistics were calculated using the custom Perl script, and all assembly sequences were contamination-checked using webtool ContEst16S to contain only a single 16S rRNA gene [35]. All assembled genomes were ensured to have ANI > 95% (using fastANI v1.1) and dDDH > 70% (using GGDC v2.1) with respect to the *C. perfringens* type strain ATCC13124 genome for the species validation [36,37]. All genomes were annotated using Prokka v1.13 with the specific *Clostridium* genus database (35 *Clostridium* species from NCBI RefSeq annotations; Appendix A) with the parameters --usegenus --mincontiglen 300 [38].

The PacBio sequencing reads of the NCTC8503 genome were retrieved from the ENA database, and the genome was reconstructed using Canu v1.5 and optimised by Circlator v1.5.5 for comparison with novel strains in this study [39,40].

### 5.6. Genomic Analysis

ARIBA was used to assemble the toxin-encoding plasmids from sequencing reads via a custom reference plasmid sequence database (at default parameters; Appendix A), and most complete sequence clusters (by sequence length) were identified for subsequent analyses [41,42]. In addition, Bandage v0.8.1 was used to investigate and extract circular sequences manually identified in the *de novo* assembly graphs generated by Spades v3.11.1 [43]. Plasmid sequences were annotated using Prokka 1.13. Linear plasmid maps were visualised using Easyfig v2.2.2 [44]. Annotated GenBank files were colour-labelled using the Artemis genome browser [45].

The toxin and antimicrobial gene identifications were performed using ABRricate v0.8.11 based on a custom toxin database (Appendix A) and the Comprehensive Antibiotic Resistance Database (CARD) v3.0.1, respectively, with options –minid = 90 and –mincov = 80 [42,46,47]. MUSCLE v3.8.31 was used to align sequences prior to the base-by-base average sequence identity (%) calculation using Panito v0.0.1 [48]. The base differences were calculated using snp-dists [49].

### 5.7. Phylogenetic Analysis

We used ParSNP v1.2 to reconstruct the Maximum Likelihood phylogenetic tree from 132 genomes (including 128 genomes batch-retrieved from the NCBI RefSeq database; July 2019; Appendix A) with the type strain ATCC13124 as the reference genome, and potential recent recombinations were removed using -x option [50,51]. The phylogenetic trees were annotated using iTOL [52]. To identify the genetic distance, snp-sites v2.3.3 and snp-dists v0.2 were using to estimate the SNP differences in the SNP alignment [49,53].

### 5.8. Data Deposition

The raw sequencing reads and genome assemblies in this study were deposited in the European Nucleotide Archive (ENA) under project PRJEB33762. An interactive annotated phylogenetic tree (Figure 2) can be viewed at https://itol.embl.de/tree/149155196252188971562847462.

## Figures and Tables

**Figure 1 toxins-11-00543-f001:**
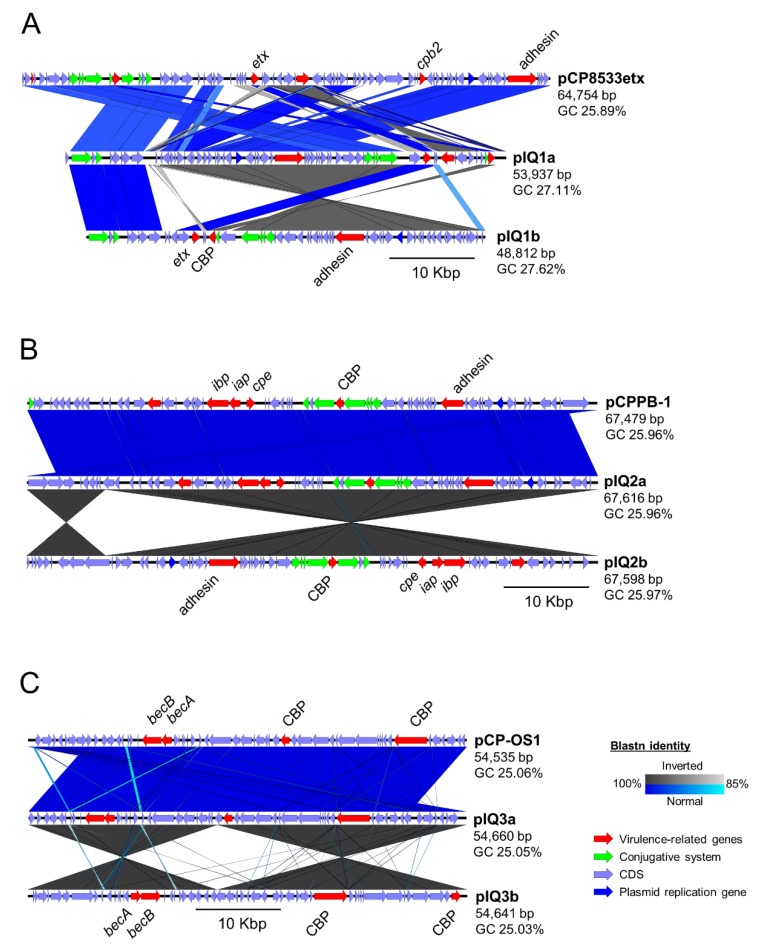
Plasmid sequence identity comparison maps showing the similarity of the reference plasmids with the computationally predicted toxin-encoding plasmids present in *C. perfringens* strains (**A**) IQ1, (**B**) IQ2, and (**C**) IQ3. In each sequence map, the reference plasmid sequence (top) was compared with the plasmid sequences reconstructed using the reference-based approach (middle) and assembly graph approach (bottom). CBP: Cell-wall Binding Protein.

**Figure 2 toxins-11-00543-f002:**
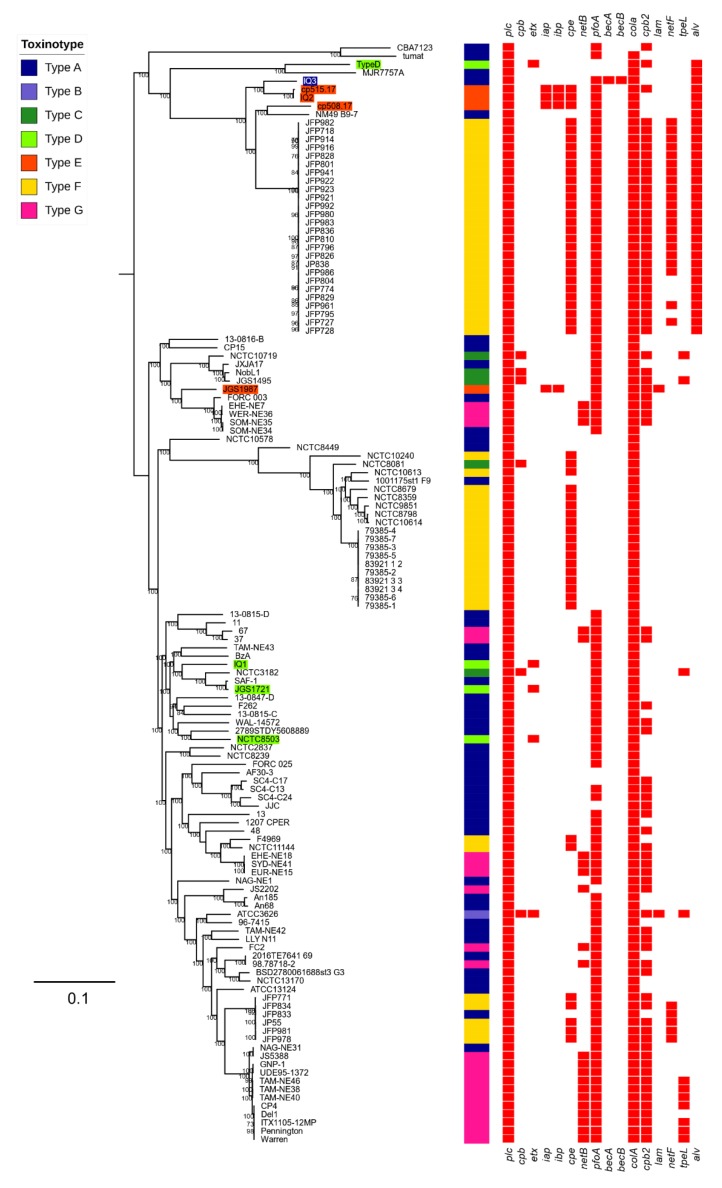
Maximum likelihood tree of 132 *C. perfringens* genomes constructed based on 115,533 SNPs aligned with the presence and absence of toxin genes and toxinotyping profiles (potential recent recombinations were removed). Bootstrap values >70% are shown on the nodes. The presence of a gene is indicated by a red cell. Toxinotype D and E, and BEC/CPILE-positive *C. perfringens* strains are colour-labelled.

**Table 1 toxins-11-00543-t001:** Genome information and statistics of three newly sequenced *C. perfringens* isolates described in this study.

Strain/Genome	*C. perfringens* IQ1	*C. perfringens* IQ2	*C. perfringens* IQ3
Genome size (bp)	3,563,782	3,438,840	3,518,141
Sequencing coverage	195×	216×	212×
Sequencing platform	Illumina HiSeq 2500	Illumina HiSeq 2500	Illumina HiSeq 2500
No. of contigs	41	15	20
Genes	3320	3199	3299
CDS	3219	3097	3196
tRNAs	91	92	92
N50 length (bp)	452,560	2,192,929	2,217,274
GC (%)	27.98	28.03	28.00
ANI (%) ^1^	98.35	96.89	97.01
dDDH (%) ^1^	87.40	74.50	75.30
Toxinotype	D	E	A (BEC-positive)
ENA sample accession	SAMEA5818795	SAMEA5818796	SAMEA5818797
ENA assembly accession	GCA_902459455	GCA_902459435	GCA_902459425

^1^ The Average Nucleotide Identity (ANI) and digital DNA-DNA Hybridisation (dDDH) were calculated with respect to the *C. perfringens* type strain ATCC13124 genome for species validation at a minimum cutoff of 95% and 70%, respectively.

**Table 2 toxins-11-00543-t002:** Nucleotide sequence similarity comparison between multiple epsilon-toxin gene *etx* variants extracted from the toxinotype B and D *C. perfringens* genomes available in the NCBI databases.

Strain	Type	SNP	Sequence Similarity (%)				
		Pos.726	ATCC3626	JGS1721	NCTC8503	IQ1	Type D
ATCC3626	B	A	100.00	99.90	99.90	99.90	100.00
JGS1721	D	G		100.00	100.00	100.00	99.90
NCTC8503	D	G			100.00	100.00	99.90
IQ1	D	G				100.00	99.90
Type D	D	A					100.00

**Table 3 toxins-11-00543-t003:** Nucleotide sequence similarity pair-wise comparison between multiple iota-toxin genes *iap* and *ibp* variants extracted from the various toxinotype E *C. perfringens* genomes.

Strain	*iap* Sequence Similarity (%)				*ibp* Sequence Similarity (%)			
	cp508.17	cp515.17	JGS1987	IQ2	cp508.17	cp515.17	JGS1987	IQ2
cp508.17	100.00	99.85	90.40	99.85	100.00	99.96	88.44	99.96
cp515.17	-	100.00	90.40	100.00	-	100.00	88.48	100.00
JGS1987	-	-	100.00	90.40	-	-	100.00	88.48
IQ2	-	-	-	100.00	-	-	-	100.00

**Table 4 toxins-11-00543-t004:** Nucleotide sequence similarity pair-wise comparison between multiple thiol-activated toxins including perfringolysin O gene *pfoA* variants and the alveolysin gene *alv* (from strain IQ2 and IQ3) computationally extracted from various *C. perfringens* genomes.

Strain	Gene	Average Nucleotide Sequence Similarity (%)					
		ATCC13124	IQ1	IQ2	IQ3	IQ2	IQ3
		*pfoA*	*pfoA*	*pfoA*	*pfoA*	*alv*	*alv*
ATCC13124	*pfoA*	100.00	98.87	97.54	97.34	86.62	86.82
IQ1	*pfoA*	-	100.00	97.34	97.14	86.62	86.62
IQ2	*pfoA*	-	-	100.00	99.67	86.62	86.62
IQ3	*pfoA*	-	-	-	100.00	86.36	86.56
IQ2	*alv*	-	-	-	-	100.00	99.43
IQ3	*alv*	-	-	-	-	-	100.00

**Table 5 toxins-11-00543-t005:** Comparisons between the reference plasmid sequences and plasmids constructed in silico using both the reference-based assembly (RBA) and assembly-graph (AG) approaches. REF: Reference sequences.

Strain	Plasmid	Method	Type	Size (bp)	Contig	GC (%)	CDS	Toxin Gene Encoded
IQ1	pIQ1a	RBA	D	53,937	13	27.11	61	*etx*
	pIQ1b	AG	D	48,812	2	27.62	58	*etx*
NCTC8533	pCP8533etx	REF	B	64,753	1	25.89	80	*etx, cpb2*
IQ2	pIQ2a	RBA	E	67,616	1	25.96	66	*iap, ibp, cpe*
	pIQ2b	AG	E	67,598	1	25.97	66	*iap, ibp, cpe*
PB-1	pCPPB-1	REF	E	67,479	1	25.96	69	*iap, ibp, cpe*
IQ3	pIQ3a	RBA	A	54,460	1	25.05	53	*becA, becB*
	pIQ3b	AG	A	54,641	1	25.03	53	*becA, becB*
OS-1	pCP_OS1	REF	A	54,535	1	25.06	54	*becA, becB*

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
