# Peer review of "Genomic Analysis of Clostridium perfringens BEC/CPILE-Positive, Toxinotype D and E Strains Isolated from Healthy Children"

_toxins, 2019, doi:10.3390/toxins11090543_

Round 1

Reviewer 1 Report

The authors have analyzed genomically the Clostridium perfringens isolates of healthy children. Eventhough the research design of this study is appropriate, some additional experiments should be performed such the qRT-PCR of toxin genes pfoA, alv and cpe to know their levels of transcription.

Also, the antimicrobial susceptibilities profiles of the C. perfringens isolates are missed in this work.

Author Response

This study represents a focused genomic analysis of C. perfringens isolates which was an invited paper for the special issue ‘omics techniques for toxins research’, thus we feel gene expression studies are out with the scope of the current work (particularly with the 3 day revision time imposed by the journal). We have highlighted these studies should form part of next stage phenotypic analysis within the discussion. With regards to antimicrobial profiling, we have performed additional analysis; profiling our three novel isolates using the CARD database (only 1 tetA(P) gene was detected), which is now highlighted within the results, with the bioinformatic tools utilised added to the methods section.

In addition, we have double-checked the manuscript to correct various spelling and grammatical errors.

Reviewer 2 Report

The manuscript on genomic analysis of Clostridium perfringens is interesting and very informative with all the detailed genetic back ground . The manuscript data helps to understand the genetic similarities between symptomatic and asymptomatic isolates. The whole genome sequencing of isolates from healthy individuals showed to carry toxin encoding genes which later under different stress conditions can get expressed to cause disease with notable symptoms. The manuscript helps in giving an clear insight about the extensive similarities showed by the toxin encoding plasmids carried by previous outbreak isolates and the ones identified from healthy individuals. These kind of studies will help in future  to deal with Clostridium perfringens infection issues.

Author Response

We thank the reviewer for their positive comments.

In addition, we have double-checked the manuscript to correct various spelling and grammatical errors.